# Red Blood Cell Fatty Acid Patterns and Cognitive Functions in Adolescents: A Pooled Analyses with Two Cohort Study Data Sets

**DOI:** 10.3390/nu17213483

**Published:** 2025-11-05

**Authors:** Nicolas Ayala-Aldana, Ariadna Pinar-Martí, Marina Ruiz-Rivera, Iolanda Lázaro, Aleix Sala-Vila, Darren R. Healy, Oren Contreras-Rodriguez, Jordi Casanova, Nuria Sola-Valls, Martine Vrijheid, Jordi Julvez

**Affiliations:** 1Clinical and Epidemiological Neuroscience (NeuroÈpia), Institut de Investigació Sanitària Pere Virgili (IISPV), 43204 Reus, Spain; ariadna.pinar@iispv.cat (A.P.-M.); marina.ruiz@iispv.cat (M.R.-R.); jordi.casanova@urv.cat (J.C.); nuria.sola@salutsantjoan.cat (N.S.-V.); 2Barcelona Institute for Global Health (ISGlobal), 08003 Barcelona, Spain; martine.vrijheid@isglobal.org; 3Faculty of Medicine and Health Science, Hospital Clinic, University of Barcelona, 08036 Barcelona, Spain; 4Departament de Ciències Experimentals i de la Salut, Facultat de Medicina i Ciències de la Vida, Universitat Pompeu Fabra (UPF), 08002 Barcelona, Spain; 5Cardiovascular Risk and Nutrition, Hospital del Mar, Medical Research Institute, 08028 Barcelona, Spain; ilazaro@researchmar.net (I.L.); asala3@imim.es (A.S.-V.); 6Centro de Investigación Biomédica en Red de la Fisiopatología de la Obesidad y Nutrición (CIBEROBN), Instituto de Salud Carlos III, 28029 Madrid, Spain; 7The Fatty Acid Research Institute, Sioux Falls, SD 57106, USA; 8Institute of Public Health and Clinical Nutrition, School of Medicine, Faculty of Health Sciences, University of Eastern Finland, 70210 Kuopio, Finland; darren.healy@uef.fi; 9Department of Psychiatry and Forensic Medicine, Autonomous University of Barcelona, 08193 Barcelona, Spain; oren.contreras@uab.cat; 10Centro de Investigación Biomédica en Red Salud Mental (CIBERSAM), Instituto de Salud Carlos III, 28029 Madrid, Spain; 11Hospital Universitari Sant Joan, 43204 Reus, Spain; 12CIBER Epidemiología y Salud Pública, Instituto de Salud Carlos III, 28029 Madrid, Spain; 13Human Nutrition Unit, Facultat de Medicina i Ciències de la Salut, Universitat Rovira i Virgili, 43204 Reus, Spain

**Keywords:** adolescent health, cognition, fatty acids, principal component analysis

## Abstract

**Objective:** Fatty acids (FAs) play a pivotal role in brain development and cognitive functions during adolescence. We aimed to investigate the association of red blood cell (RBC) FA patterns and several high order neuropsychological functions in adolescents. **Methods:** The study followed a cross-sectional design. Principal component analysis was applied to 22 FA species previously measured in RBC membranes (exposure variable) to identify FA principal components (PCs) from two cohorts of adolescents in Catalonia, Spain (mean age = 14.53 years). Multiple linear regression was then used to examine associations between PC FAs and cognitive outcomes—working memory, fluid intelligence, and risky decision-making (gain and loss domains). Regression models were adjusted for child sex, age, body mass index, maternal education, and cohort enrollment. **Results:** Three FA PCs (eigenvalues > 2.0) were retained for the current study: a very-long chain FAs PC, a long-chain omega-6 FA PC and an omega-3 FA PC. The omega-3 FA PC showed a positive association with scores of fluid intelligence ($\beta_{1}$ = 0.14, CI = 0.05, 0.24, *p* for trend = 0.003) and risky decision-making (loss domain) ($\beta_{1}$ = 0.27, CI = 0.03, 0.52, *p* for trend = 0.030). The very-long chain FAs and long-chain omega-6 FAs patterns showed no significant associations with any cognitive outcome. The PC of omega-3 FA and fluid intelligence associations remained significant after multiple testing corrections. **Conclusions:** After applying an agnostic approach of multiple FAs in RBC, we found omega-3 FA patterns were positively associated with fluid intelligence among adolescents.

## 1. Introduction

Adolescence is marked by significant changes in cognitive functions, influenced by diet, education, and life experiences [1]. During this period, the development of critical cognitive functions like working memory, fluid intelligence, and decision-making becomes pivotal for everyday tasks, academic achievement, and problem-solving [2,3,4]. These neuropsychological functions are primarily mediated by the prefrontal cortex of the brain [5], whose synaptic functions and homeostatic balance are influenced by fatty acids (FAs) [6]. Docosahexaenoic acid (DHA) is an omega-3 FA that rapidly accrues in the prefrontal cortex during the first two decades of life, accounting for ∼15% of the total FA composition in adulthood [7]. This prompted the interest in whether sustained consumption of foods rich in DHA and other omega-3 FAs might help in promoting child and adolescent mental and brain health [8]. However, DHA is only one of many diet- and metabolically derived FAs that could play a role in brain function [9]. Exposure to these FAs is best determined by analysis of blood (specifically of red blood cell, RBC) FA profiles as opposed to relying on biased and imprecise data derived from dietary questionnaires [10]. The use of these objective FA biomarkers (which can be influenced by diet, metabolism, genetics, or hormones) in observational studies is increasingly regarded as a tool to help in the designing of new trials on diet and cognition in adolescents [11]. The main advantage of using this methodology is that it captures the FA composition of RBC membranes, which reflects both habitual dietary intake and endogenous metabolism over a medium-term period (approximately two to three months). This approach provides a more stable and representative measure of long-term FA exposure compared to plasma measurements [12], which are more susceptible to short-term fluctuations due to recent dietary intake and postprandial changes.

In the adolescent population, a higher RBC omega-3 index (DHA + EPA) was associated with enhanced information processing and selective attention [13] whereas in individuals with attention-deficit/hyperactivity disorder, lower levels have been linked to poorer cognitive performance and more behavioral symptoms [14,15]. In the adult population, a higher omega-3 index of RBC was associated with larger hippocampal volumes and better abstract reasoning, with similar associations observed for DHA and EPA [16]. From a clinical perspective, adults with Alzheimer’s and a low omega-3 index were cross-sectionally linked to poorer performance on the Wechsler Memory Scale and higher tau accumulation among carriers of the apolipoprotein E ε4 allele [17]. Membrane FAs in RBC may represent a sensitive biomarker of cognitive health across the lifespan.

Due to the correlation among FAs, isolating the ‘pure effect’ of each individual FA is difficult, making it challenging to investigate the relationship between FA patterns and cognitive functions in the adolescent population. Sophisticated approaches for determining FA patterns include principal components analysis (PCA), a type of dimensionality reduction technique [18,19]. In this perspective, limited research exists on the collective impact of different FA species on cognitive functions in adolescents. We found only one study, conducted in a healthy elderly population (n = 386), using a PCA technique. In that study, higher omega-3 FAs and the omega-3 principal component (PC) were positively associated with better nonverbal memory and processing speed [20]. However, comparable studies in adolescent populations are scarce. Most research to date has focused on single FA or broader dietary patterns [8], rather than examining the combined effects of multiple FA species using multivariate approaches such as PCA. This gap limits our understanding of how complex FA profiles may influence cognitive development during adolescence—a period of rapid neurodevelopment and heightened nutritional sensitivity. Consequently, evidence from younger populations remains largely extrapolated from adult or elderly cohorts, underscoring the need for dedicated studies in adolescents.

FA metabolism may be involved in structural and functional changes in the brain during adolescence. A key challenge is to study the different combinations of FA patterns on adolescent cognitive function. In the current study, we hypothesized that omega-3 RBC FA patterns would relate to better cognitive functions in an adolescent population. To address this challenge, we conducted a PCA in order to identify the correlational structure of twenty-two RBC FAs, and analyze which combinations of FAs are most strongly associated with cognitive performance.

## 2. Material and Methods

### 2.1. Study Design and Participants

The current study is based on a cross-sectional design using baseline information from the Walnuts Smart Snack Dietary Intervention Trial (WSS) and the 14th year follow-up visit of the Spanish birth cohort Childhood and Environment (INfancia y Medio Ambiente, INMA) Project. On the one hand, WSS was developed to determine if dietary supplementation with 30 g of raw walnut kernels per day for six months resulted in beneficial improvements in cognitive and socioemotional development when compared to a control group of healthy adolescents [21]. A total of 771 participants from 11 high schools were recruited over a year (2015–2016). Participants completed several computer-based neuropsychological tests and provided information on their lifestyle and dietary preferences prior to randomization. All information about the clinical trial procedure is described in the WSS protocol [22]. The trial study received permission from Parc Salut Mar’s Clinical Research Ethics Committee (approval number: 2015/6026/I). For the current study, we used the WSS baseline data, a subsample with biological sampling that included adolescents with a mean age of 13.81 years old (n = 332).

On the other hand, the INMA study is a multi-centric research project carried out in several Spanish cities. The project aims to analyze the relationship between prenatal and postnatal environmental factors, and child health development. Extensive assessments were conducted on 3100 pregnant women and their children, including all the Spanish cities of the project. Data were collected through physical examinations, questionnaires, interviews, ultrasound, and biological samples. Over the years, the group of children from the city of Sabadell has been evaluated approximately every 2 years: at 6 and 14 months, and at 2, 4, 7, 9, 11, and 14 years of age. Detailed information is described in the INMA protocol [23]. INMA cohort received permission from “Instituto Municipal de Asistencia Sanitaria” Research Ethic Committee (approval number: 2005/2106/I). For the current research, we gathered data on the population sample from Sabadell city (INMA-Sabadell cohort, n = 657 at pregnancy recruitment). We used participant information data from the cohort’s 14th–16th year of follow up (n = 328).

### 2.2. Fatty Acid Determination

For both cohorts, WSS and INMA-Sabadell, a nurse collected blood samples following an overnight fast. Subsequently, we centrifuged the samples at 2500 g for 20 min at 20 °C within 4 h of extraction. Packed RBCs were preserved at –80 °C until the FAs were analyzed. The FA profile within the RBCs was determined using gas chromatography, coupled to either a flame ionization detector (WSS) or electron ionization mass spectrometry (INMA), as described in references [24] and [25], respectively. In both cases, the quantification of each FA was expressed as a percentage of the 22 FAs identified in the sample. The list and descriptive statistics of total FAs measured in the current study are provided in Appendix A.

### 2.3. Primary Outcomes

To assess working memory in both cohorts, we utilized the computer-based n-back test. The participants were instructed to indicate whenever a current stimulus matched the one presented “n” trials earlier (1-, 2-, 3-, and 4-back) [26]. We collected direct metrics such as hits, correct rejections, false alarms, and misses for each trial, along with hit reaction time. Overall accuracy, comprising both, hits and correct rejections, was calculated, as well as d prime (d′), for each block individually. Calculation of d′ involved the following formula: d′ = z (hit rate) − z (false alarm rate). A higher d′ value suggests enhanced detection capability and, consequently, more precise performance. We used d′ 4-back tasks as measure of high order working-memory function (outcome) in posterior multivariate regression models.

To assess fluid intelligence in the WSS cohort, we utilized the computer-based Primary Mental Ability test (PMA) [27]. We employed the reasoning test (PMA-R), which entails selecting a letter from a set of six possible alternatives to complete a given sequence of letters, thereby evaluating inductive reasoning based on letter patterns. The total score corresponds to the number of correct item responses, with higher scores indicating stronger fluid intelligence abilities. To assess fluid intelligence in the INMA-Sabadell cohort, the computer-based Raven’s Standard Progressive Matrices (Raven) was used. It is similar to PMA-R but uses a series of different geometric designs, formed by non-verbal elements. The Raven test consists of 60 items, with the total number of correct responses serving as the outcome measure [28]. A higher Raven test score reflects superior fluid intelligence, which pertains to the capacity to solve novel reasoning problems with minimal reliance on prior learning. In order to analyze fluid intelligence as a common variable, we combined the PMA-R and Raven tests by standardizing their results using z-scores (mean = 0, standard deviation = 1).

To assess decision-making function in both cohorts, we used the computer-based Roulette task. This task assesses whether participants adjust their propensity for risk in accordance with the probabilities and significance of potential outcomes [29]. The Roulette’s task is essentially a chance-based game with the objective of maximizing monetary gains. Two roulette wheels are displayed on the screen, each divided into segments containing varying amounts of money. The left roulette features one segment with a larger sum (USD 2), while the remaining segments are empty (USD 0). Conversely, all segments on the right roulette offer the same amount (USD 1). The left roulette represents a “risky” choice, as the outcome is uncertain, and the likelihood of landing on a higher amount diminishes with more segments. In contrast, the right roulette is a “safe” option, guaranteeing a USD 1 outcome. Moreover, the roulettes can be colored red or blue, with red indicating a loss and blue signifying a gain. Participants must decide which roulette to spin multiple times, adjusting their risky behavior based on outcome probabilities and significance. Scores are evaluated as total risk adjustment to gauge risk sensitivity, where a risk adjustment index closer to 0 suggests greater risk insensitivity. The roulette Task in Risk Adjustment for Gain (CUPRAG) and roulette Task Risk Adjustment for Loss (CUPRAL) scores were evaluated as primary outcomes using multiple linear regression models. Participants were to win as much money as possible in the Gain domain (CUPRAG) and to lose as little money as possible in the Loss domain (CUPRAL). Both outcomes’ scores range from −9 to 9, with higher scores indicating better performance. Scores below 0 (meaning more risky decisions when the options are unfavorable) are atypical and could suggest random responses or a misunderstanding of the task.

### 2.4. Reduction in Dimensionality of Fatty Acids: Principal Component Analysis

We conducted a PCA on the RBC FAs of the adolescents from the WSS and INMA-Sabadell cohorts. We analyzed all FAs measured in both cohorts (Appendix A). The researchers did not perform any selection of FAs, nor did they select FAs for convenience, given the correlated associations of FAs in RBC (Appendix A). After PCA analyses, PC with an eigenvalue > 2.0 were retained. Subsequently, a varimax rotation was applied to the retained components to facilitate the variable contribution inside each component, followed by an analysis of loading scores within each component. We described the components using variable loadings greater than |0.2| and conducted a detailed analysis of the variables with the highest loading scores. Additionally, standardized scores for each adolescent within each component were calculated as an “exposure variable” for posterior regression analysis. Higher levels of standardized scores indicate greater representation of the FAs with positive contributions to the component. Conversely, lower levels of standardized scores indicate greater representation of the FAs with negative contributions to the component. Moreover, the standardized scores were converted into tertiles to avoid potential problems with non-linear relationships between the exposure and the outcome in the multivariate regression. To assess the batch effect, PCA was performed separately for the two cohorts, WSS and INMA-Sabadell, to explore potential batch effects and the distribution of samples in the reduced-dimensional space. First, the dataset was split by cohort, retaining the first 22 variables for each group. PCA was then applied to each cohort independently using standardized and centered variables. Unrotated PCA scores for the first two components were extracted for each cohort, and a new variable indicating cohort membership was added. The scores of the first two PCs were then combined into a single dataset to allow visualization of cohort separation. The mean explained variance of PC1, PC2, and PC3 across the two cohorts was calculated to quantify the proportion of total variance captured. The combined scores were plotted using a scatterplot of PC1 versus PC2, PC1 versus PC3, and PC2 versus PC3 with points colored by cohort and confidence ellipses overlaid to illustrate the spread and potential batch effects. To further evaluate the stability and reproducibility of the PCA, we performed Procrustes analysis. Procrustes analysis was applied to the PCA loadings (WSS as reference, INMA-Sabadell as target) to quantitatively assess structural similarity. The primary metric was the Procrustes Root Mean Squared Error (RMSE), which quantifies the average distance between corresponding loadings after optimal superimposition; a lower RMSE indicates higher structural consistency between the cohorts’ underlying structures. The analysis also calculated the scaling factor, which represents the magnitude by which the target structure was scaled for the best fit; a factor close to 1.0 indicates similar overall structure size and variance.

### 2.5. Statistical Analysis

Associations between retained FA components and cognitive scores (“working memory”, “fluid intelligence” and “risky decision-making”) were evaluated using multivariable linear regression models. FA components were evaluated as ordinal variables (tertiles) computed from their standardized scores (continuous exposure variable). For all analyses, fully adjusted models were assessed with mandatory covariates, including age (continuous), biological sex (dichotomous), and Body Mass Index (BMI, continuous) in the main regression model. BMI was computed using the reference population according to the World Health Organization recommendations, and the calculation was adjusted by sex and age to provide a BMI z-score. Additionally, maternal education level (university studies/lower education) and cohort of enrollment (WSS/INMA-Sabadell) were used as mandatory confounders in the main regression model. Most statistical analyses, with the exception of basic descriptive analyses, were exclusive to adolescents with complete information on the biological, outcome, and covariate variables included in the final regression models (maximum n = 578).

For all regression models, a *p*-value of <0.05 was considered statistically significant. Specifically, in the analysis of the ordinal exposure variable (categories 1, 2, and 3), we considered a *p* for trend below 0.05 as threshold of statistical significance. We did not conduct an in-depth analysis when significance was found only in the categorical exposure variable without the presence of a significant *p* for trend in ordinal tertiles (categories 1, 2, and 3). To analyze multiple test corrections, we conducted the False Discovery Rate (FDR) test. Sensitivity analyses were performed to assess the confounding effect on exposure variables (PC of FAs) using the same sample as the main models, considering that adolescent cognition can be influenced by multiple factors, including individual, family, and environmental characteristics. We analyzed the effect of physical activity, adherence to the Mediterranean Diet (MedDiet), maternal mental disorder, and maternal social class. Physical activity of participants was categorized into three levels: “sedentary to low,” “moderate,” and “active to quite active”. Adherence to the MedDiet was assessed using the KIDMED questionnaire (continuous score). Maternal mental disorder was recorded as “no” or “yes” levels. For the WSS cohort, it was obtained from a self-reported questionnaire asking about a previous clinical diagnosis of a mental health disorder, and for INMA-Sabadell, it was obtained through the Symptom Checklist-90-Revised (SCL-90), using a threshold of 63 on the Global Severity Index to indicate psychological distress as a proxy of mental disorder. Finally, maternal social class was recorded as “working class,” “medium class,” and “medium-high to high class”. A change in the beta coefficient of ±10% or more was considered an important modification of the size effect of the exposure variable and its associations with FA PC. An interaction model (PC FAs × Cohort) is evaluated to determine whether the cohort—with its distinct measurement methodology—directly influences the effect of the exposure on the outcome. Statistical analyses were conducted using an R base (version 4.4.0) and R Studio (version 4.2.3). Finally, the library “zscorer” (version 0.3.1) was used to obtain the BMI-z score [30], and the libraries “factoextra” (version 1.0.7) and “psych” (version 2.4.3) were employed in PCA [31,32].

## 3. Results

The baseline characteristics of the study population are shown in Table 1. Cognitive performance significantly differed between the WSS and INMA-Sabadell cohorts. For working memory, assessed with the N-back 4 (d’), adolescents from INMA-Sabadell obtained slightly higher scores compared to those from WSS (mean = 2.48, SD = 1.22 vs. mean = 1.74, SD = 1.11; *p*-value < 0.001). Similar patterns were observed for risky decision-making in the Roulettes Task. Specifically, INMA-Sabadell participants scored higher in both the CUPRAG (mean = 5.25, SD = 2.42 vs. mean = 4.53, SD = 2.42; *p*-value < 0.001) and CUPRAL domains (mean = 5.16, SD = 2.21 vs. mean = 3.98, SD = 2.63; *p*-value < 0.001). Regarding covariates, participants had an equal gender distribution, but ages differed between the WSS (mean = 13.81, SD = 0.92) and INMA-Sabadell (mean = 15.36, SD = 0.76) cohorts (*p*-value < 0.001). In the WSS cohort, the majority of participants’ mothers had a university-level education (65%), whereas only 35% of the mothers of the INMA-Sabadell participants had the same level of education.

Three PCs were extracted using PCA; eigenvalue > 2.0 criteria for PC retention and loadings greater than |0.4| were statistically relevant for the pattern analysis. The three PCs retained accounted for 57.20% of the total variance for the FA data (Appendix A) according Kaiser criteria. No additional relevant information was added in the subsequent PCs, as confirmed by the scree plot (Appendix A).

Factor loadings for the FA patterns are presented as their loading contributions in Figure 1. PC1 was named very-long chain FAs because it was mainly characterized by C24:1 *n*-9 (nervonic acid), C22:0 (behenic acid) and C24:0 (lignoceric acid), ordered from largest to smallest loading. Both C16:1 *n*-7 (palmitoleic acid) and C18:0 (stearic acid) showed negative loadings. PC2 was named long-chain omega-6 because it was characterized by C20:4 *n*-6 (arachidonic acid, AA), C22:4 *n*-6 (adrenic acid), and C20:2 *n*-6 (eicosadienoic acid, EDA) as positive loadings. The FAs with negative loadings were C18:1 *n*-9 *cis* (oleic acid), C16:0 (palmitic acid), C14:0 (myristic acid) and all-*trans* C18:1. Lastly, PC3 was named omega-3 FAs because it was characterized mainly by positive loadings of C20:5 *n*-3 (eicosapentaenoic acid, EPA), C22:6 *n*-3 (DHA), and C22:5 *n*-3 (docosapentaenoic acid, DPA) The FAs with negative loadings were C22:4 *n*-6 (adrenic acid) and C22:5 *n*-6 (docosapentaenoic acid omega-6). The summary of the variables’ loadings for each component (PC1, PC2, and PC3) is also displayed in the PCA biplot (Appendix A). PCA was initially performed to visualize potential cohort differences (batch effects) between the WSS and INMA-Sabadell datasets using unrotated PC scores (PC1 vs. PC2, PC1 vs. PC3, and PC2 vs. PC3). While the cohorts were separated by color, the subsequent analysis support that this separation did not represent a major structural difference between the datasets. Procrustes analysis was applied to the PCA loadings to quantify the structural agreement between the cohorts. The overall agreement was moderate and tolerable, as indicated by a low Procrustes RMSE of 0.294 (IQR: 0.21, 0.34) for the combined 3D structure. The optimal transformation required a Scaling Factor of 0.59 for the INMA-Sabadell cohort. The largest residual distances, which highlight minor differences, were concentrated in specific FAs: C14:0 (distance: 0.40), followed by the long-chain *n*-3 PUFAs: C22:6 *n*-3 (DHA) (distance: 0.32) and C20:5 *n*-3 (EPA) (distance: 0.26). The overall moderate RMSE, however, confirms that these were localized differences within an otherwise consistent structural representation (Appendix A).

Eigenvalue criteria > 2.0. The variance (%) of each component is presented as follows: PC1 (33.76%), PC2 (13.23%), and PC3 (10.22%). The total variance explained, by these three principal components, is 57.20%.

PC1, very long-chain FAs; PC2, long-chain omega-6 FAs; PC3, omega-3 FAs.

Table 2 presents the fully adjusted associations between FA patterns and working memory and fluid intelligence outcomes in the adolescents’ population. For fluid intelligence, PC3 omega-3 FAs showed a significant linear trend across tertiles ($\beta_{1}$ = 0.14, CI = 0.05, 0.24, *p* for trend = 0.003), with a positive association in both, the 2nd tertile ($\beta_{1}$= 0.25, CI = 0.07, 0.44, *p*-value = 0.007) and 3rd tertile ($\beta_{1}$ = 0.29, CI = 0.10, 0.47, *p*-value = 0.003) compared to the 1st tertile of FAs content. There were no statistically significant associations for any FA pattern and working memory (N-back d4′).

Table 3 presents the fully adjusted model associations between FA patterns and roulette task outcomes in the adolescents’ population. The PC3 omega-3 FAs showed a positive association with CUPRAL across tertiles ($\beta_{1}$ = 0.27, 95% CI  = 0.03, 0.52, *p* for trend = 0.030), with a higher score in the 3rd tertile ($\beta_{1}$ = 0.54, 95% CI  = 0.05, 1.04, *p*-value = 0.030), compared to the 1st tertile of FAs. There were no statistically significant associations between any FA pattern and CUPRAG.

After correcting *p*-values for multiple testing using the FDR method (Appendix A), statistical significance was retained only for the following association (*p*-value < q-value): PC3 omega 3 FAs and fluid intelligence (*p*-value= 0.003, q-value= 0.004), providing strong evidence for this association.

In sensitivity analyses (Appendix A), adjusting for MedDiet and socioeconomic class slightly reduced the association between PC 3 omega-3 FAs and fluid intelligence (by <10%), but it remained statistically significant. No relevant changes were observed when the main model was additionally adjusted by physical activity or maternal mental disorder. No changes were observed either when including the cohort as an interaction term (Appendix A). 

## 4. Discussion

In this cross-sectional study, we used data from adolescents in the WSS and INMA-Sabadell cohorts to identify complex correlation relationships among RBC FAs and cognitive functions through PCA. Our results show significant associations between PC3 omega-3 FAs and cognitive outcomes, including fluid intelligence and risky decision-making. After FDR correction, the PC3 omega-3 FAs pattern and fluid intelligence showed a strong association. No statistically significant association was found between PC1 very-long chain FAs, PC2 long-chain omega-6 FAs, and cognitive functions.

Regarding PC1 very-long chain pattern, this PC was not associated with working memory, fluid intelligence, or risky decision-making. The PC1 very-long chain pattern is primarily characterized by C24:1 *n*-9 (nervonic acid), C22:0 (behenic acid), and C24:0 (lignoceric acid). Nervonic acid has been identified as an important FA for nervous system development and the myelination process [33]. Behenic acid and lignoceric acid have been associated with promoting healthy aging and preventing adverse outcomes, including mental health diseases [34]. No studies have been found examining these FAs as both exposure and cognitive outcomes. Future research should be conducted to clarify their role during adolescence.

Although we did not observe significant associations between working memory and any of the FA patterns, a previous clinical trial in a child and adolescent population (6 to 12 years) suggested that fish oil consumption (DHA supplementation) enhanced attention, memory, and cognitive processing, as evidenced by changes in brain activity during working and long-term memory tasks [35]. Similar associations are found when EPA + DHA is used as supplementation in the adolescent population [36]. Furthermore, when older age groups were studied, similar results were observed. A clinical trial showed that DHA supplementation enhanced memory in healthy young adults (aged 15 to 45) with low habitual DHA intake [37].

Our results revealed non-significant associations between PC2 long-chain omega-6 scores and cognitive functions such as working memory, fluid intelligence, and risky decision-making scores. EDA and AA are the two omega-6 PUFAs with higher loadings in PC2 long-chain omega-6 FA. Previous research has shown that EDA and AA have been involved in pro-inflammatory processes [38,39]. A cross-sectional study with children (7 to 9 years old) revealed that those with lower omega-6 to omega-3 ratios exhibited shorter initial processing times on the spatial working memory task and reduced average planning times [40]. Recent evidence suggests discontinuing the use of the omega-6/omega-3 PUFA ratio, as it wrongly assumes all omega-6 PUFAs are harmful and all omega-3 PUFAs counteract their effects [41]. However, there is also controversial information supporting the neuroprotective functions of AA in human population [9,42]. In our case, obtaining neutral results may be due to several factors that deserve further exploration. First, the long-chain omega-6 cluster did not incorporate C18:2 *n*-6 (linoleic acid, LA), which is an essential FA that is abundant in RBC membranes. Because LA can be elongated and desaturated to obtain long-chain omega-6 FAs, there has been much controversy on whether dietary LA might have deleterious effects [43]. Second, the loss of linearity in the regression may also be factors that mask the effect of long-chain omega-6 FAs. Finally, more studies should be conducted to clarify the role of long-chain omega-6 FAs in the adolescent population.

The non-significant associations observed for PC1 very long-chain FAs and PC2 long-chain omega-6 FAs may be partly explained by several factors. Measurement variability in FA quantification could have introduced noise, attenuating potential effects. In addition, multicollinearity among FAs—given their shared metabolic pathways and correlated dietary sources—may have contributed to overlapping variance structures within the PCs, thereby diluting specific associations. These aspects highlight the complexity of interpreting PCA-derived components in biological data and suggest that larger cohorts might better capture subtle relationships between FA patterns and the cognitive outcomes. On the other hand, we found positive associations between fluid intelligence and risky decision-making with PC3 omega-3 FAs. The PC3 pattern is characterized by omega-3 FAs, namely EPA, DHA, and DPA. These results add to the increasing evidence that these omega-3 FAs might improve cognitive functions. Diving into bimolecular mechanisms, EPA has a neuroprotective role, blocking the neuroendocrine and cognitive effects of interleukin 1β [44]. DHA and DHA-derived molecules modulate an anti-inflammatory response, enhancing neuroprotection (via neuroprotectin D1) and promoting brain cell survival and repair [45]. DPA can, through oral administration, provide neuroprotective effects, reducing sphingomyelinase and caspase-3 activities, oxidative stress, and microglial activation in aged rats, all of which are linked to improvements in long-term potentiation and spatial memory [46]. Additionally, ALA promotes the biosynthesis of anti-inflammatory mediators (i.e., lipoxins in the central nervous system) and can be converted to EPA and DHA, although this conversion occurs in limited quantities in human metabolism [47]. A clinical trial conducted by Widenhorn-Müller et al. reported that supplementation with 720 mg/day of *n*-3 PUFAs increased EPA and DHA concentrations in RBC membranes and improved working memory performance, while no significant effects were found for other cognitive outcomes or for parent- and teacher-rated behaviors (age 6–12 years) [48]. As for decision-making, another previous clinical trial with a healthy population (age = 18.6 to 25.8) showed that the group supplemented with fish oil containing 2.3 g of omega-3 PUFA (1.74 g EPA, 0.25 g DHA) exhibited fewer risk-averse decisions compared to the placebo group (olive oil) [49]. Moreover, a systematic review found that in studies involving typically developing children, daily supplementation of 450 mg or more of DHA and EPA led to improvements in overall cognitive function in half of the investigations (i.e., decision-making, attention, and reaction time) [36]. Thus, consistent with our findings and based on data from children and adolescents, omega-3 FAs may play a cognitive protective effect, particularly on fluid intelligence and risky decision-making. Previous research using mediation analysis reported that a specific omega-3 PUFA pattern from PCA, comprising ALA, stearidonic, and eicosatrienoic acids, was associated with fluid intelligence, with total gray matter volume of the left frontoparietal cortex fully mediating this relationship [50]. These findings are consistent with our results, although the original study was conducted in an elderly population. Taken together, omega-3 FAs, particularly those long-chain varieties, appear to support cognitive function across the lifespan. Our findings are particularly relevant for adolescents, a period characterized by ongoing maturation of the prefrontal cortex, hippocampus, and other cortical and subcortical regions involved in executive function, learning, and decision-making. These brain areas are particularly rich in omega-3 FAs, which play a crucial role in synaptic plasticity, membrane fluidity, and neuronal signaling.

An important strength of this study lies in the use of lipidomic techniques to directly assess FA levels in RBCs, offering greater accuracy and sensitivity compared to other dietary assessment methods, such as food frequency questionnaires. RBCs have a long lifespan of approximately 120 days, making their FA profile a more accurate indicator of long-term dietary intake compared to total plasma or serum [51]. Another important strength is the use of computerized neuropsychological tests to measure cognitive functions, rather than relying on more generalized indicators (e.g., school tests or self-reported rating scales). We adopted a non-supervised machine learning approach (PCA) to assess the impact of multiple FAs on the cognitive function of adolescents, which provides better methodological robustness. Finally, our findings are strengthened by the fact that the associations between FA profiles and cognitive outcomes remained stable after sensitivity analyses accounting for physical activity, adherence to a MedDiet, maternal mental health, and socioeconomic status. Nonetheless, the current study also faced some limitations. The main limitation is its cross-sectional nature, which makes it impossible to establish causality due to the temporal ambiguity inherent in capturing data at a single point in time. Another limitation is compounded by the potential for selection bias, as the sample may not be representative of the broader population, thereby affecting the generalizability of the results. Besides that, observational data may have the possibility of residual confounding, since there may be unmeasured environmental or lifestyle factors (e.g., sleep patterns, stress levels, family environment, exposure to toxins, genetic background) that influence both, FA RBC levels and cognitive function. Moreover, the lack of longitudinal data further restricts insights into the progression and dynamics of the studied FAs and cognitive functions. However, the use of two cohorts allowed us to obtain robust results regarding the associations between FA factors and cognitive functions in adolescents. Conversely, we only used the overlapping information from the WSS and INMA-Sabadell cohorts, which may lead to a loss of data regarding the participants’ cognitive characteristics, exposure, and confounders in the study. Regarding the computer cognitive tests, the N-back, PMA, Raven test, and Roulette Task assess specific aspects of cognition and may not capture overall cognitive functioning. Task performance could also be influenced by factors such as participants’ motivation, fatigue, attention, behavioral symptoms, or prior experience with similar tasks, which were not directly controlled. Finally, after applying corrections for multiple testing, the statistical significance of the association between PC3 omega-3 FAs and risky decision-making was lost. However, we emphasize the relevance of this relationship, given the supporting scientific evidence discussed earlier. Moreover, focusing solely on controlling Type I errors could actually increase the risk of Type II errors, potentially overlooking meaningful associations [52].

## 5. Conclusions

Overall, our findings suggest that omega-3 FAs (EPA, DHA, and DPA altogether) of the RBC membrane had a positive association with fluid intelligence in the adolescent population from Catalonia (Spain). However, the interpretation of the association with risky decision-making should be made with caution, as it did not remain significant after FDR correction. Additionally, no association was observed between “very-long chain FAs” or “long-chain omega-6 FAs” with working memory, fluid intelligence, or risky decision-making. Future research should provide further insight into those FAs holistically as potential targets to promote cognitive health, particularly within adolescent populations. Given the cross-sectional nature of our findings, causal relationships cannot be established, and longitudinal or intervention studies are needed to confirm these associations and better understand their potential impact on cognitive development.

## Figures and Tables

**Figure 1 nutrients-17-03483-f001:**
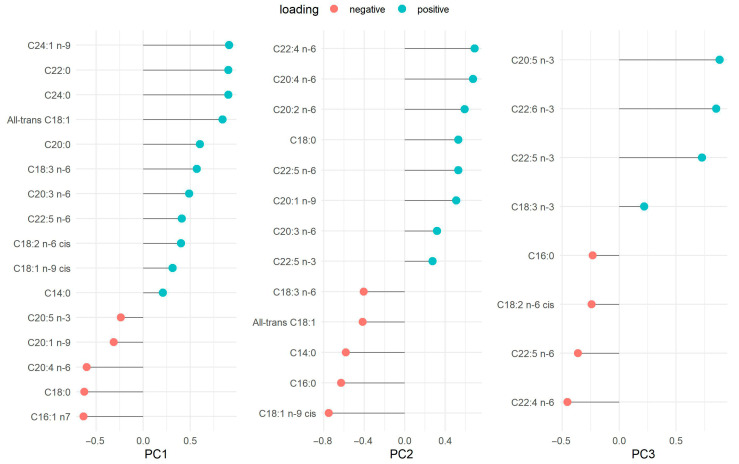
Fatty acid loadings in principal component analysis.

**Table 1 nutrients-17-03483-t001:** Baseline characteristics of the study population.

Variable	N	Total ^a^ N = 660	WSS ^a^ N = 332	INMA-Sabadell ^a^ N = 328	*p*-Value ^b^
N-back 4 (d prime)	605	2.09 (1.22)	1.74 (1.11)	2.48 (1.22)	<0.001
PMA-R test (total of correct responses) ^c^	758		16.21 (5.79)		
Raven test (total correct responses) ^c^	777			35.53 (4.49)	
Roulettes Task (CUPRAG)	608	4.87 (2.45)	4.53 (2.42)	5.25 (2.42)	<0.001
Roulettes Task (CUPRAL)	608	4.54 (2.51)	3.98 (2.63)	5.16 (2.21)	<0.001
Sex	660				0.874
Female		336 (51%)	168 (51%)	168 (51%)	
Male		324 (49%)	164 (49%)	160 (49%)	
Age (years)	621	14.53 (1.15)	13.81 (0.92)	15.36 (0.76)	<0.001
Height (cm)	656	164.58 (8.77)	162.21 (8.69)	166.95 (8.21)	<0.001
Weight (kg)	654	57.84 (13.41)	53.85 (11.66)	61.87 (13.87)	<0.001
BMI z-score	615	0.34 (1.11)	0.27 (1.05)	0.42 (1.17)	0.164
Physical activity	636				0.001
Sedentary to low		150 (24%)	56 (18%)	94 (29%)	
Moderate		155 (24%)	70 (23%)	85 (26%)	
Active to quite active		331 (52%)	184 (59%)	147 (45%)	
Adherence to the MedDiet	630	6.24 (2.34)	6.85 (2.02)	5.67 (2.47)	<0.001
Mother studies	620				<0.001
Secondary school or less		305 (49%)	121 (37%)	184 (64%)	
University		315 (51%)	210 (63%)	105 (36%)	
Maternal mental disorder	612				0.840
No		529 (86%)	274 (87%)	255 (86%)	
Yes		83 (14%)	42 (13%)	41 (14%)	
Social class	574				<0.001
Working		95 (17%)	36 (13%)	57 (20%)	
Middle		279 (49%)	105 (37%)	174 (60%)	
Middle high to high		200 (35%)	143 (50%)	59 (20%)	

WSS, Walnuts Smart Snack Intervention Trial; INMA, “Infancia y Medio Ambiente”; PMA, Primary Mental Abilities in WSS cohort; CUPRAG roulettes task risk adjustment—gain domain; CUPRAL roulettes task risk adjustment—Loss domain. ^a^ Mean and standard deviation (SD) for continuous variables and number of participants and percentage (%) for categorical variables. ^b^ Wilcoxon rank sum test for continuous variables and Pearson’s chi-squared test for categorical variables. ^c^ Maximum sample size of cohort used to compute the posterior z-score for fluid intelligence outcome.

**Table 2 nutrients-17-03483-t002:** Multiple linear regression models for fatty acids in principal component and cognitive function (working memory and fluid intelligence) in the adolescent population.

		Working Memory 4—Back (d Prime)				Fluid Intelligence PMA-R and Raven		
**Variable**	**n**	$\beta_{1}$ ** ^a^ **	**95% CI**	***p*-Value**	**n**	$\beta_{1}$ ** ^a^ **	**95% CI**	***p*-Value**
PC1: Very-long chain FAs								
Tertile 1	163	Ref.			156	Ref.		
Tertile 2	200	0.06	−0.23, 0.35	0.691	199	−0.12	−0.35, 0.12	0.327
Tertile 3	205	0.24	−0.15, 0.63	0.223	207	−0.12	−0.42, 0.19	0.457
Tertiles in continuous ^b^	568	0.13	−0.07, 0.32	0.203	562	−0.05	−0.20, 0.10	0.500
PC2: Long-chain omega-6 FAs								
Tertile 1	193	Ref.			191	Ref.		
Tertile 2	189	0.01	−0.23, 0.24	0.957	187	−0.10	−0.29, 0.09	0.311
Tertile 3	186	0.18	−0.08, 0.44	0.177	184	−0.13	−0.34, 0.08	0.233
Tertiles in continuous ^b^	568	0.09	−0.04, 0.22	0.180	562	−0.06	−0.17, 0.04	0.230
PC3: Omega-3 FAs								
Tertile 1	187	Ref.			184	Ref.		
Tertile 2	199	−0.12	−0.35, 0.11	0.319	195	0.25	0.07, 0.44	0.007
Tertile 3	182	0.07	−0.17, 0.31	0.575	183	0.29	0.10, 0.47	0.003
Tertiles in continuous ^b^	568	0.03	−0.09, 0.15	0.575	562	0.14	0.05, 0.24	0.003

PMA, primary mental abilities; PC, principal component; FAs, fatty acids; n, number of subjects with available data; CI, confidence interval; Ref, reference group. ^a^ Beta coefficient (slope) and 95% CI (Confidence Interval) estimated using multiple linear regression models adjusted for sex, age, maternal education, BMI z-score and cohort. Tertile 1 of exposure variable (principal component) used as reference group. ^b^ *p*-value is *p* for trend.

**Table 3 nutrients-17-03483-t003:** Multiple linear regression models for fatty acids in principal component and cognitive function (risky decision-making) in the adolescent population.

		Risky Decision-Making Roulettes Task						
		CUPRAG				CUPRAL		
**Variable**	**n**	$\beta_{1}$ ** ^a^ **	**95% CI**	***p*-Value**	**n**	$\beta_{1}$ ** ^a^ **	**95% CI**	***p*-Value**
PC1: Very-long chain FAs								
Tertile 1	164	Ref.			164	Ref.		
Tertile 2	203	−0.27	−0.85, 0.30	0.347	200	−0.10	−0.70, 0.50	0.746
Tertile 3	211	−0.47	−1.23, 0.29	0.225	208	−0.47	−1.26, 0.33	0.251
Tertiles in continuous ^b^	578	−0.23	−0.61, 0.15	0.229	572	−0.24	−0.64, 0.15	0.228
PC2: Long-chain omega-6 FAs								
Tertile 1	196	Ref.			194	Ref.		
Tertile 2	194	−0.01	−0.48, 0.46	0.958	192	−0.19	−0.68, 0.30	0.448
Tertile 3	188	−0.25	−0.76, 0.27	0.350	186	−0.43	−0.97, 0.11	0.116
Tertiles in continuous ^b^	578	−0.12	−0.38, 0.14	0.354	572	−0.22	−0.49, 0.05	0.116
PC3: Omega-3 FAs								
Tertile 1	188	Ref.			188	Ref.		
Tertile 2	201	0.08	−0.39, 0.54	0.747	199	0.21	−0.27, 0.69	0.383
Tertile 3	189	0.29	−0.18, 0.76	0.221	185	0.54	0.05, 1.04	0.030
Tertiles in continuous ^b^	578	0.15	−0.09, 0.38	0.219	572	0.27	0.03, 0.52	0.030

CUPRAG roulettes task risk adjustment—gain domain; CUPRAL roulettes task risk adjustment—loss domain; PC, principal component; FAs, fatty acids; n, number of subjects with available data; CI, confidence interval; Ref, reference group. ^a^ Beta coefficient (slope) and 95% CI (Confidence Interval) estimated using multiple linear regression models adjusted for sex, age, maternal education, BMI z-score and cohort. Tertile 1 of exposure variable (principal component) used as reference group. ^b^ *p*-value is *p* for trend.

## Data Availability

The datasets generated during and/or analyzed during the current study are available from the corresponding author on reasonable request. The datasets are not publicly available because they contain sensitive biological and sociodemographic information about cohort participants.

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
