# Peer review of "Red Blood Cell Fatty Acid Patterns and Cognitive Functions in Adolescents: A Pooled Analyses with Two Cohort Study Data Sets"

_nutrients, 2025, doi:10.3390/nu17213483_

Round 1
Reviewer 1 Report
Comments and Suggestions for Authors
This is a well-organized study with adequate novelty. Some points should only be addressed.
- The authors should try to increase a bit more the introduction section concerning the contents of FAs in RBCs and how this is related with cognitive function.
- Conclusions section should also be increase a bit.
- English language editing is recommended.
English language editing is recommended.
Author Response
Reviewer 1.
This is a well-organized study with adequate novelty. Some points should only be addressed.
- The authors should try to increase a bit more the introduction section concerning the contents of FAs in RBCs and how this is related with cognitive function.
Response: We thank the reviewer for this comment. We agree with this comment. We have increased the content related to FAs in RBCs and cognitive function across the lifespan. The update can be found in the second paragraph of the Introduction section as follows:
- Conclusions section should also be increase a bit.
Response: We thank the reviewer for this comment. We have updated all the findings of current research, including those not statistically associated with the cognitive outcomes. Additionally, we have increased the future research on FA of RBC and cognitive functions. The update can be found in the conclusion section as follows:
- English language editing is recommended.
Response: We thank the reviewer for this suggestion. The manuscript has been carefully revised and professionally edited to improve clarity, grammar, and overall readability.
Reviewer 2 Report
Comments and Suggestions for Authors
Thank you for the opportunity to review the article entitled ”Red Blood Cell Fatty Acid Patterns and Cognitive Functions in Adolescents: A Pooled Analyses With Two Cohort Study Data Sets”.
Adolescence is a very tumultuous period, and caring for psychological and physical well-being is a natural concern. In general, the article is well done, well-written and clear for any Reader. That is why specialists and any interested researcher could extract knowledge from the paper.
To be suitable for publication, I express here some concern:
- There are some similarities with previously published articles, using the same general database. The authors should clearly present this aspect; even if the populations are separated, it seems like using the database is not so totally ethical - please clarify the slice-salamy aspect.
- The authors should present the selection of respondents, because adolescence is a period characterized by a lot of psychological and emotional transformation; how the researchers made the difference between normal respondents and those having language, cognitive, or other disabilities, including psychological troubles (ex, depressive symptoms or others), which restrain the cognitive result.
- The authors should highlight the limitations regarding the use of all instruments (including computer-based evaluation methods),
- The authors should be aware that a correlation relationship is not equal to a causal relationship. In my opinion, relating fatty acids/diet to the decision-making process looks more like a forced result.
- Of course, diet is important during adolescence, but the results obtained cannot be seen as being absolute, and to minimize or reduce the importance of all other aspects of life (such as sleep, genetic background, time invested, learning abilities, motivation, attention, social background etc)
- It is not very clear the variables the authors included in the research. In the Results section is appearing the level of education of mothers. What are the variables included in the research (apart from medical and psychological data) so we can identify some for respondents and some for mothers? Are they important?
- Also, it is not fair to present the results in the Discussion section by explaining them using the results obtained by the same team in the same project. In the Discussion section, the results must be analyzed by comparing them - similar or opposite results to some other studies.
- Limitations of the study - must also refer the the exclusion of many variables that could explain the results, which were not considered for the present research.
Author Response
Reviewer 2
Thank you for the opportunity to review the article entitled “Red Blood Cell Fatty Acid Patterns and Cognitive Functions in Adolescents: A Pooled Analyses With Two Cohort Study Data Sets”.
Adolescence is a very tumultuous period, and caring for psychological and physical well-being is a natural concern. In general, the article is well done, well-written and clear for any Reader. That is why specialists and any interested researcher could extract knowledge from the paper.
To be suitable for publication, I express here some concern:
- There are some similarities with previously published articles, using the same general database. The authors should clearly present this aspect; even if the populations are separated, it seems like using the database is not so totally ethical - please clarify the slice-salamy aspect.
Response: We thank the reviewer for this comment. Below are the related articles on the web and the objectives of our article, which is different from those previously published:
“Walnuts, Long-Chain Polyunsaturated Fatty Acids, and Adolescent Brain Development: Protocol for the Walnuts Smart Snack Dietary Intervention Trial” doi: 10.3389/fped.2021.593847
This study presents the research protocol for the “Walnuts Smart Snack Intervention Trial” (WSS). It outlines the study variables and the planned analytical approach. In brief, this research paper describes a six-month, population-based randomized controlled trial involving 800 adolescents designed to evaluate the effectiveness of a daily walnut supplementation (four walnuts, equivalent to 30 g of kernels or approximately 1.5 g of ALA) in enhancing neuropsychological and socio-emotional development compared with a control group that did not receive the walnut intervention. Since this is only a protocol paper it does not show any result.
“Red blood cell omega-3 fatty acids and attention scores in healthy adolescents” doi: 10.1007/s00787-022-02064-w
This research is a cross-sectional research of 372 participants from WSS cohort. This study examined the associations between docosahexaenoic acid (DHA) and alpha-linolenic acid (ALA) and attention function among a healthy young population using linear regression models. This study only includes three omega-3 fatty acids.
“Effect of walnut consumption on neuropsychological development in healthy adolescents: a multi-school randomised controlled trial” doi: 10.1016/j.eclinm.2023.101954
This research is a clinical trial involving 771 healthy adolescents aged 11–16 years, who were randomly assigned to either an intervention or a control group. The study conducted a six-month, multi-school, randomized controlled nutritional intervention to assess whether walnut consumption has beneficial effects on the neuropsychological and behavioral development of adolescents.
Given the context of the previously mentioned studies, along with their objectives and methodologies, the present research—titled “Red Blood Cell Fatty Acid Patterns and Cognitive Functions in Adolescents: A Pooled Analysis Using Two Cohort Study Data Sets”—aims to investigate the association between 22 fatty acids and cognitive performance in adolescents. The study combines data from two cohorts: the Walnuts Smart Snack (WSS) cohort (n = 332) and the “Infancia y Medio Ambiente” (INMA)–Sabadell cohort (n = 328). In other words, we have nearly doubled the sample size compared to previous studies to prove our hypothesis of a wider sample size. Methodologically, we employed principal component analysis (PCA) to investigate the combined effect of fatty acids on cognition. We employed a non-supervised machine learning technique to create “latent variables” of fatty acids involved in red blood cell membranes, as metabolomics biomarkers.
Previous cross-sectional studies, in scientific literature, have typically assessed the individual effects of specific fatty acids such as ALA or DHA. However, there is limited evidence on the combined influence of fatty acids,(22 different fatty acids in our current study in submission) or on how the relationship between omega-3 fatty acids and other short- or long-chain fatty acids may impact cognitive functions during adolescence.
- The authors should present the selection of respondents, because adolescence is a period characterized by a lot of psychological and emotional transformation; how the researchers made the difference between normal respondents and those having language, cognitive, or other disabilities, including psychological troubles (ex, depressive symptoms or others), which restrain the cognitive result.
Response: We thank the reviewer for this important comment. We have now included a description of the selection of participants in the manuscript and have acknowledged as a study limitation that adolescents with language, cognitive, or other psychological difficulties—including depressive symptoms—might influence cognitive outcomes. Sensitivity analyses were conducted to evaluate potential environmental effects; however, it was not possible to fully control for these factors because the INMA Sabadell cohort did not have information on behavioral symptoms. These limitations have been explicitly addressed in the Discussion section as follows:
“Regarding to computer cognitive test, the N-back, Primary Mental Abilities, Raven test, and Roulette Task assess specific aspects of cognition and may not capture overall cog-nitive functioning. Task performance could also be influenced by factors such as par-ticipants’ motivation, fatigue, attention, behavioral symptoms, or prior experience with similar tasks, which were not directly controlled”.
- The authors should highlight the limitations regarding the use of all instruments (including computer-based evaluation methods).
Response: We thank the reviewer for this comment. The limitations regarding computerized instrument was updated in the limitations of discussion section as follows:
“Regarding to computer cognitive test, the N-back, Primary Mental Abilities, Raven test, and Roulette Task assess specific aspects of cognition and may not capture overall cognitive functioning. Task performance could also be influenced by factors such as par-ticipants’ motivation, fatigue, attention, behavioral symptoms, or prior experience with similar tasks, which were not directly controlled”.
- The authors should be aware that a correlation relationship is not equal to a causal relationship. In my opinion, relating fatty acids/diet to the decision-making process looks more like a forced result.
Response: We thank the reviewer for this comment. The entire manuscript was carefully reviewed to ensure the correct use of language and to refer only to associations, given the limitations of the cross-sectional design. Regarding the association between Omega-3 PC and the risky decision-making outcome, although this association is statistically significant, its interpretation should be made with caution due to the FDR adjustment. However, the models were compared with sensitivity analyses, and we did not observe any confounding effects. The sensitivity analyses are presented in Supplementary Tables 5 and 6.
- Of course, diet is important during adolescence, but the results obtained cannot be seen as being absolute, and to minimize or reduce the importance of all other aspects of life (such as sleep, genetic background, time invested, learning abilities, motivation, attention, social background etc)
Response: We thank the reviewer for this comment. We acknowledge that several individual and social factors may influence the observed associations. To address this concern, we conducted additional sensitivity analyses including variables that could potentially affect the effect size of the exposure, such as physical activity, adherence to the Mediterranean Diet, maternal mental disorder, and maternal social class. The inclusion of these covariates did not substantially change the magnitude or direction of the associations, suggesting that these factors did not act as confounders in our models. This consistency across models reinforces the robustness of the observed associations.
- It is not very clear the variables the authors included in the research. In the Results section is appearing the level of education of mothers. What are the variables included in the research (apart from medical and psychological data) so we can identify some for respondents and some for mothers? Are they important?
Response: We thank the reviewer for this comment. In addition to medical and psychological data, our study included a set of sociodemographic, lifestyle, and maternal variables. For the children/adolescents (respondents), we included age, sex, physical activity, adherence to the Mediterranean Diet, and cognitive performance measures. For the mothers, we included maternal educational level, maternal social class, and maternal mental health status. These variables were considered relevant as they may influence both the exposure and the outcomes, and were therefore included either as covariates in the main analyses or in sensitivity analyses to assess the robustness of the associations. The inclusion of these factors allows for a more comprehensive understanding of the observed relationships and helps reduce potential confounding effects.
- Also, it is not fair to present the results in the Discussion section by explaining them using the results obtained by the same team in the same project. In the Discussion section, the results must be analyzed by comparing them - similar or opposite results to some other studies.
Response: We thanks the reviewer for this comment. In response, we have removed the previously reported results from our research group to avoid redundancy and ensure clarity. Additionally, we have revised and expanded the discussion section to more thoroughly address the potential effect modification of omega-3 supplementation on red blood cell (RBC) membrane composition and its implications for cognitive functioning. These revisions aim to provide a clearer interpretation of our findings within the context of existing evidence:
“A clinical trial conducted by Widenhorn-Müller et al. reported that supplementation with 720 mg/day of n-3 PUFAs increased EPA and DHA concentrations in RBC mem-branes and improved working memory performance, while no significant effects were found for other cognitive outcomes or for parent- and teacher-rated behaviors (age 6-12 years)”.
- Limitations of the study - must also refer the the exclusion of many variables that could explain the results, which were not considered for the present research.
Response: We thank the reviewer for this comment. We highlighted the limitation of residual confounding in our study, as there may exist variables relevant to the estimation of RBC FAs and adolescent cognition, such as sleep patterns, stress levels, family environment, exposure to toxins, or genetic background. The updated text is located in the Limitations subsection of the Discussion section as follows:
“Besides that, observational data may have the possibility of residual confounding, since there may be unmeasured environmental or lifestyle factors (e.g., sleep patterns, stress levels, family environment, exposure to toxins, genetic background) that influence both, FA RBC levels and cognitive function”.
Reviewer 3 Report
Comments and Suggestions for Authors
Dear Authors,
Thank you fo rproviding me with the opportunity to review this interesting paper. Below, I have listed my comments:
1) The paper should make clearer why using PCA to identify FA patterns in adolescents is methodologically or biologically innovative. For example, is this the first study linking multivariate FA profiles (vs. individual FAs) to multiple cognitive domains (working memory, fluid intelligence, decision making)? Explicitly stating this would highlight the paper’s originality.
2) It’s not entirely clear why and how two different cohorts (WSS and INMA) were merged. Are there potential batch effects or methodological differences (e.g., FA quantification methods) that could affect comparability? You could include a sentence justifying the decision to pool data and also describe any harmonization procedures or sensitivity analyses.
3) In the discussion, the non-significant results (e.g., for PC1 and PC2) are well-described but not critically analyzed. The authors might briefly explore whether sample size, measurement variability, or multicollinearity among FAs could have influenced the null findings.
4) The conclusion could benefit from explicitly reiterating the cross-sectional nature of the findings and the need for longitudinal or intervention studies.
5) The discussion sometimes reads as a literature summary rather than a critical interpretation of the present findings. The authors could better highlight why their results matter specifically for adolescents, a population with unique neurodevelopmental processes.
I hope this feedback is helpful.
Author Response
Reviewer 3
- The paper should make clearer why using PCA to identify FA patterns in adolescents is methodologically or biologically innovative. For example, is this the first study linking multivariate FA profiles (vs. individual FAs) to multiple cognitive domains (working memory, fluid intelligence, decision making)? Explicitly stating this would highlight the paper’s originality.
Response: We thank the reviewer for this comment. We agree with the suggestion and have expanded the point of the “lack of research literature” in the Introduction by highlighting the relevance of studying combined FA profiles and cognitive domains in adolescent populations. The statement has been updated as follows:
“In this perspective, limited research exists on the collective impact of different FA species on cognitive functions in adolescents. We found only one study, conducted in a healthy elderly population (n= 386), using a PCA technique. In that study, higher ome-ga-3 FAs and the omega-3 principal component were positively associated with better nonverbal memory and processing speed (14). However, comparable studies in adoles-cent populations are scarce. Most research to date has focused on single fatty acids or broader dietary patterns, rather than examining the combined effects of multiple FA species using multivariate approaches such as PCA. This gap limits our understanding of how complex FA profiles may influence cognitive development during adolescence—a period of rapid neurodevelopment and heightened nutritional sensitivity. Consequently, evidence from younger populations remains largely extrapolated from adult or elderly cohorts, underscoring the need for dedicated studies in adolescents”.
- It’s not entirely clear why and how two different cohorts (WSS and INMA) were merged. Are there potential batch effects or methodological differences (e.g., FA quantification methods) that could affect comparability? You could include a sentence justifying the decision to pool data and also describe any harmonization procedures or sensitivity analyses.
Response: We thank the reviewer for this important observation. The two cohorts (WSS and INMA) were combined to increase statistical power and improve the generalizability of the findings. Fatty acid (FA) data from both cohorts were expressed as percentages of total fatty acids, ensuring comparability across datasets. Before performing the PCA, all FA variables were centered and scaled, as required by the R function used. Visual inspections of potential batch effects and principal component overlap were conducted, indicating good alignment between cohorts. In addition, a Procrustes analysis showed moderate and stable consistency, supporting the reproducibility of the PCA structure across cohorts. Finally, an interaction term was included in the models to evaluate whether the cohort variable introduced substantial differences in exposure, which was not the case. Please, review the updated Methods Section, Results (Supplementary Table 5-7, Supplementary Figure 4)
- In the discussion, the non-significant results (e.g., for PC1 and PC2) are well-described but not critically analyzed. The authors might briefly explore whether sample size, measurement variability, or multicollinearity among FAs could have influenced the null findings.
Response: We thank the reviewer for this valuable comment. We have updated the Discussion section to provide a more critical interpretation of the non-significant associations observed for PC1 (very long-chain FAs) and PC2 (long-chain omega-6 FAs). The revised text now considers that measurement variability in FA quantification and multicollinearity among FAs—due to shared metabolic pathways and correlated dietary sources—may have attenuated potential effects and contributed to overlapping variance within the PCs. These changes remained stable after conducting sensitivity analyses, reinforcing the robustness of our observations. The updated discussion section aims to better contextualize the challenges of interpreting PCA-derived components in biological data and highlights that larger cohorts may be required to detect subtle relationships between FA patterns and cognitive outcomes.
“The non-significant associations observed for PC1very long chain FAs and PC2 long-chain omega-6 FAs may be partly explained by several factors. Measurement variability in FA quantification could have introduced noise, attenuating potential effects. In addition, multicollinearity among FAs—given their shared metabolic pathways and correlated dietary sources—may have contributed to overlapping variance structures within the PCs, thereby diluting specific associations. These aspects highlight the complexity of interpreting PCA-derived components in biological data and suggest that larger cohorts might better capture subtle relationships between FA patterns and the cognitive outcomes"
- The conclusion could benefit from explicitly reiterating the cross-sectional nature of the findings and the need for longitudinal or intervention studies.
Response: We thank the reviewer for this helpful comment. The Conclusion section has been revised to explicitly emphasize the cross-sectional nature of our findings and to highlight the need for longitudinal and intervention studies to further explore the observed associations. The updates in the Conclusion section can be found as follows:
“Overall, our findings suggest that omega-3 FAs (EPA, DHA and DPA altogether) of RBC membrane had a positive association on fluid intelligence and risky decision-making. No association was observed between “very-long chain FAs” or “long-chain omega-6 FAs” with working memory, fluid intelligence, or risky decision-making. Future research should provide further insight into those FAs holistically as potential targets to promote cognitive health, particularly within adolescent populations. Given the cross-sectional nature of our findings, causal relationships cannot be established, and longitudinal or intervention studies are needed to confirm these associations and better understand their potential impact on cognitive development”.
- The discussion sometimes reads as a literature summary rather than a critical interpretation of the present findings. The authors could better highlight why their results matter specifically foradolescents, a population with unique neurodevelopmental processes.
Response: We thank the reviewer for this constructive comment. The Discussion section has been updated to provide a more critical interpretation of our findings, emphasizing their specific relevance for adolescents, a population characterized by unique neurodevelopmental processes. Please, review the updated version of Discussion Section.
Reviewer 4 Report
Comments and Suggestions for Authors
Overview of the manuscript
This study investigates the relationship between red blood cell (RBC) fatty acid (FA) composition and cognitive functions among adolescents. Using data from two Spanish cohorts—the Walnuts Smart Snack (WSS) Trial and the INMA-Sabadell birth cohort to identify FA patterns and examine their associations with working memory, fluid intelligence, and decision-making abilities. The authors use blood samples analysis for 22 RBC FAs using gas chromatography, correlation with cognitive functions were assessed via standardized computer-based tasks and complex statistical methods. The authors find that the omega-3 FAs PC showed a positive association with scores of fluid intelligence and risky decision making, whereas the very-long chain FAs and long-chain omega 6 FAs patterns showed no significant associations with any cognitive outcome. The authors conclude highlighting that omega-3 FAs of red blood cell membrane had a protective association on fluid intelligence and risky decision-making, inviting future research to provide further findings on FA as potential targets to promote cognitive health, particularly within adolescent populations.
GENERAL COMMENT
This is a well-designed and presented study. The statistically rigor applied contributes to provide valuable evidence to the field of nutritional neuroscience, especially regarding adolescent brain development. The methodology is detailed and rigorous. The employ of principal component analysis (PCA) to identify correlated FA patterns and subsequent regression analyses with covariate adjustments is methodologically adequate and give particular strength to the results. The work opportunely highlights the methodological strengths and limitations.
Specific comments
Pag. 2, line 62-64: the issue of exposure to FAs best determined by analysis of red blood cells should be better explained and emphasised.
The minor methodological differences between cohorts should be better evidenced.
Author Response
Reviewer 4
Overview of the manuscript
This study investigates the relationship between red blood cell (RBC) fatty acid (FA) composition and cognitive functions among adolescents. Using data from two Spanish cohorts—the Walnuts Smart Snack (WSS) Trial and the INMA-Sabadell birth cohort to identify FA patterns and examine their associations with working memory, fluid intelligence, and decision-making abilities. The authors use blood samples analysis for 22 RBC FAs using gas chromatography, correlation with cognitive functions were assessed via standardized computer-based tasks and complex statistical methods. The authors find that the omega-3 FAs PC showed a positive association with scores of fluid intelligence and risky decision making, whereas the very-long chain FAs and long-chain omega 6 FAs patterns showed no significant associations with any cognitive outcome. The authors conclude highlighting that omega-3 FAs of red blood cell membrane had a protective association on fluid intelligence and risky decision-making, inviting future research to provide further findings on FA as potential targets to promote cognitive health, particularly within adolescent populations.
General Comment.
This is a well-designed and presented study. The statistically rigor applied contributes to provide valuable evidence to the field of nutritional neuroscience, especially regarding adolescent brain development. The methodology is detailed and rigorous. The employ of principal component analysis (PCA) to identify correlated FA patterns and subsequent regression analyses with covariate adjustments is methodologically adequate and give particular strength to the results. The work opportunely highlights the methodological strengths and limitations.
Specific comments
- Pag. 2, line 62-64: the issue of exposure to FAs best determined by analysis of red blood cells should be better explained and emphasised.
Response: We thank the reviewer comment. We agree with the suggestion, and the text has been revised accordingly to better explain and emphasize the importance of assessing fatty acid exposure through red blood cell analysis. These changes have been made in the Introduction section, specifically in the lines indicated by the reviewer:
“The main advantage of using this methodology is that it captures the FA composition of RBC membranes, which reflects both habitual dietary intake and endogenous metabolism over a medium-term period (approximately two to three months). This ap-proach provides a more stable and representative measure of long-term FA exposure compared to plasma measurements , which are more susceptible to short-term fluctua-tions due to recent dietary intake and postprandial changes”.
- The minor methodological differences between cohorts should be better evidenced.
Response: We thank the reviewer for this important observation. The two cohorts (WSS and INMA) were combined to increase statistical power and improve the generalizability of the findings. Fatty acid (FA) data from both cohorts were expressed as percentages of total fatty acids, ensuring comparability across datasets. Before performing the PCA, all FA variables were centered and scaled, as required by the R function used. Visual inspections of potential batch effects and principal component overlap were conducted, indicating good alignment between cohorts. In addition, a Procrustes analysis showed moderate and stable consistency, supporting the reproducibility of the PCA structure across cohorts. Finally, an interaction term was included in the models to evaluate whether the cohort variable introduced substantial differences in exposure, which was not the case. Please, review the updated Methods Section, Results Section (Supplementary Table 5-7, Supplementary Figure 4).
Round 2
Reviewer 3 Report
Comments and Suggestions for Authors
Thank you, Authors, for revising the paper. Good luck with the rest of the process.
Author Response
Thank you, Authors, for revising the paper. Good luck with the rest of the process
Response: We thank the reviewer for the valuable comments that improved the scientific quality throughout the manuscript.